# Effect of High-Temperature Annealing on Graphene with Nickel Contacts

**Tommi Kaplas [1],\*, Vytautas Jakstas [1], Andrius Biciunas [1], Algimantas Luksa [2], Arunas Setkus [2], Gediminas Niaura [3] and Irmantas Kasalynas [1]**

[1] Department of Optoelectronics, Center for Physical Sciences and Technology, Sauletekio Ave. 3, LT-10257 Vilnius, Lithuania; vytautas.jakstas@ftmc.lt (V.J.); andrius.biciunas@ftmc.lt (A.B.); irmantas.kasalynas@ftmc.lt (I.K.)

[2] Department of Physical Technologies, Center for Physical Sciences and Technology, Sauletekio Ave. 3, LT-10257 Vilnius, Lithuania; algimantas.luksa@ftmc.lt (A.L.); arunas.setkus@ftmc.lt (A.S.)

[3] Department of Organic Chemistry, Center for Physical Sciences and Technology, Sauletekio Ave. 3, LT-10257 Vilnius, Lithuania; gediminas.niaura@ftmc.lt

\* Correspondence: tommi.kaplas@ftmc.lt

**Abstract:** Graphene has shown great potential for ultra-high frequency electronics. However, using graphene in electronic devices creates a requirement for electrodes with low contact resistance. Thermal annealing is sometimes used to improve the performance of contact electrodes. However, high-temperature annealing may introduce additional doping or defects to graphene. Moreover, an extensive increase in temperature may damage electrodes by destroying the metal–graphene contact. In this work, we studied the effect of high-temperature annealing on graphene and nickel–graphene contacts. Annealing was done in the temperature range of 200–800 °C and the effect of the annealing temperature was observed by two and four-point probe resistance measurements and by Raman spectroscopy. We observed that the annealing of a graphene sample above 300 °C increased the level of doping, but did not always improve electrical contacts. Above 600 °C, the nickel–graphene contact started to degrade, while graphene survived even higher process temperatures.

**Keywords:** graphene; annealing; doping; electric contacts

## 1. Introduction

In the past 15 years, graphene has attracted an enormous amount of interest from the scientific community. Unique properties like ballistic, high-mobility charge carriers at room temperature have made this monoatomic layer of graphite a material with high potential for futuristic, high-speed electronic applications. However, in order to implement graphene for electronic devices, there is a requirement to introduce low-resistance ohmic contacts to the graphene–metal interface.

A lot of work has been dedicated to finding good electrical contacts for graphene and other 2D materials [1–4]. For this purpose, many metals (e.g., Ag, Al, Au, Co, Cr, Cu, Fe, Ni, Pd, Pt, and Ti) have been tested [2–7]. Among these metals, nickel has especially proven to have great potential, since it creates a good contact with graphene due to the chemisorption mechanism [8]. Furthermore, the thermal annealing of samples has been shown to enhance the electric contact between nickel and graphene [9–12].

In addition, to enhance electrical contacts, high-temperature annealing is frequently used for cleaning graphene after depositing it on a dielectric surface [13]. More specifically, graphene is conventionally synthetized on a transition metal substrate by chemical vapor deposition and then transferred by a polymer support onto a dielectric substrate. The transfer process usually

contaminates the surface of graphene with residual polymer particles, but it is possible to remove the polymer contaminants from the graphene surface by thermal annealing in order to improve its performance [13,14].

Graphene cleaning and contact annealing at elevated temperatures poses the question of what annealing does to the properties of graphene and at what temperature annealing should be performed. A precise understanding of what happens to graphene during annealing is crucially important, and therefore, this topic has been widely investigated [10,12–16]. For instance, Lin et al. observed that rather low-temperature (200–300 °C) thermal annealing is a simple technique to clean polymer particles from the surface of graphene [13]. However, thermal cleaning leaves a fair amount of polymer residuals on the surface of graphene, and more importantly, it also increases its doping level after exposure to air and humidity [13,15,16]. Moreover, Cheng et al. reported that an optimized annealing process might increase electron mobility in graphene [12]. They found that annealing graphene at 200 °C almost doubled electron mobility. In addition, they observed that, when the annealing temperature increases above 200 °C, the electron/hole mobility starts to decrease. Furthermore, Leong et al. explained the mechanism of how annealing affects Ni–graphene contacts at 300 °C [10]. They found that annealing graphene with a nickel contact causes some carbon atoms from graphene to be dissolved into Ni, resulting in small holes in the graphene lattice. Eventually, this creates strong chemical bonding between nickel and graphene, which in turn improves the performance of the electrode [10]. Therefore, based on these earlier studies, one may conclude that the most beneficial thermal annealing temperature range is 200–300 °C.

While there have been several studies focusing on graphene annealing with and without contacts below 400 °C, only a small number of studies indicate what happens to graphene above 400 °C [15]. Moreover, there is still no clear understanding of what happens to graphene in contacts at very high temperatures. In this work, we extend previous works by systematically studying the effect of extreme annealing temperature on graphene and nickel–graphene contacts. In our experiment, we annealed a graphene sample with nickel contacts in the temperature range of 200−800 °C. Changes in graphene properties were observed by measuring its electrical resistivity and Raman spectra.

## 2. Results

### 2.1. Sample Preparation

Graphene was grown on a copper foil by a conventional hot wall chemical vapor deposition (CVD) technique [17]. The synthetized graphene film was transferred by the wet transfer method on an oxidized (280 nm) silicon substrate [18,19]. Briefly, the graphene sample was spin-coated with a poly(methyl methacrylate) (PMMA) film of 100 nm thickness. The underlying copper was etched in a mild ferric chloride solution (~5 g FeCl + 100 mL $H_2O$) overnight. After etching of the copper substrate, the graphene+PMMA was cleaned with an RCA-SC2 solution (10 mL HCl + 10 mL $H_2O_2$ + 150 mL $H_2O$) for 10 min. Thereafter, the graphene+PMMA sample was rinsed twice in water for 20 min and deposited onto the silicon substrate. The sample was dried at room temperature overnight, and finally, the PMMA support was removed by using an acetone bath (2 h), followed by rinsing in isopropanol and water for 5 and 10 min, respectively.

Nickel contacts were fabricated by an electron beam evaporator through a shadow mask. Figure 1 schematically illustrates the geometry of the sample. Disk-shaped contacts were 500, 600, and 750 μm in diameter (d1, d2, and d3, respectively). The contact spacing was 2 mm, and the thickness of the evaporated Ni layer was 50 nm. Ni–graphene contact was expected to be ohmic [7], which we confirmed by measuring a linear I–V curve.

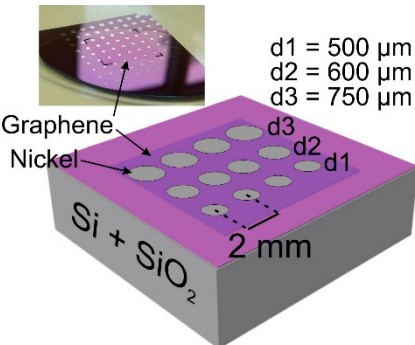

**Figure 1.** A schematic illustration of the graphene with nickel contacts on the oxidized silicon substrate. The thickness of the contact nickel film was 50 nm, and the spacing between contacts was 2 mm. The inset shows a photograph of the actual sample.

The sample was annealed using a rapid thermal annealing device (Unitemp RTV-100HV) under an argon atmosphere (in 2000 sccm flow at atmospheric pressure). Before the contact annealing, the sample was pre-annealed at 200 °C for 3 min to remove moisture from the sample surface. Thereafter, the process temperature was increased up to the actual annealing temperature at a ramp-up rate of 10 °C/s. The sample was annealed for 3 min at the actual annealing temperature and then cooled down to 200 °C in Ar atmosphere. To increase the cooling rate, the rest of the cooling from 200 to 100 °C was done under nitrogen atmosphere. At 100 °C, the sample was removed from the chamber to room temperature.

## 2.2. Electrical Characterization

Before and after annealing, the resistance of the sample was measured by the two and four-point probe techniques (by Keithley 2400 SMU Source Measure Unit connected to the probe station). The four-point-probe technique diminishes the effect of the contact resistance, and the resistance of graphene (Rg) was obtained directly from the experimental data [20]. The two-point-probe technique measures the total resistance (Rtot) [20], including the resistance of the two contact points (Rc) and Rg (see Figure 2a). When two contacts are identical, Rc can be estimated by subtracting Rg from Rtot and dividing this value by two (see Figure 2c).

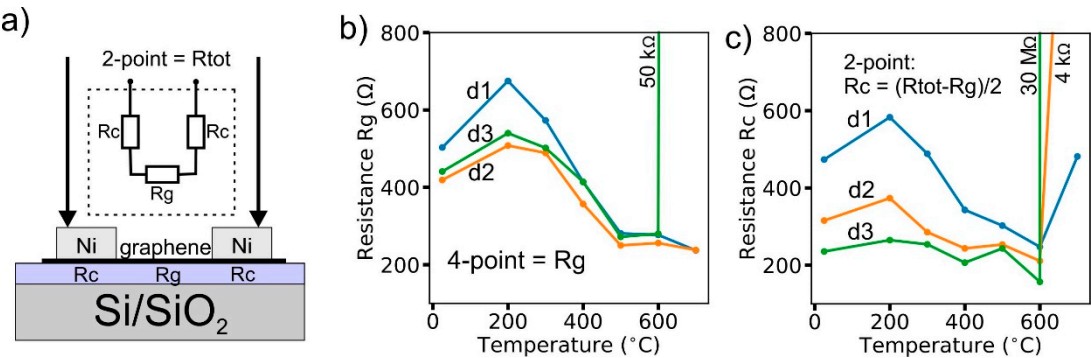

**Figure 2.** Electrical characterization of the sample. (**a**) A schematic drawing of the two-point measurement with an equivalent circuit of the sample. The equivalent circuit has two equal resistors for contacts (Rc) and one resistor for graphene (Rg). (**b**) Dependence of graphene resistance on the annealing temperature. Rg decreased significantly when the annealing temperature increased above 300 °C. (**c**) Dependence of contact resistance on the annealing temperature. For the smallest contacts (d1), the annealing decreased the contact resistance before it broke down above 600 °C. However, the contact resistance became less dependent on the annealing temperature in the case of a larger contact size (d2 and d3) due to limitations of the setup used for contact resistance measurements. Rtot: total resistance.

Figure 2b,c shows the resistance of graphene and contacts, respectively, after annealing. Figure 2b shows that the Rg value of the sample annealed in the temperature range of 200–300 °C was comparable to that of the un-annealed sample. However, annealing of the sample at temperatures above 300 °C decreased Rg significantly. When the annealing temperature was increased up to 500 °C, Rg saturated to 260 ± 20 Ω.

Moreover, Figure 2c shows that the contact between graphene and nickel started to degrade at above 600 °C and thus Rc increased rapidly. Moreover, the size of the contact seems to play an important role. For the smallest contact (d1), the annealing improved the electrical contact, which was seen as a decrease in the Rc value. However, the annealing procedure seemed to have only minor effects on the middle and largest size contacts (d2 and d3, respectively).

### 2.3. Raman Characterization

Raman spectroscopy is a powerful tool to probe the properties of graphene. In our experiment, we measured the Raman spectra of the annealed graphene sample using the 532 nm excitation wavelength in a Renishaw inVia Raman microscope. Spectra were measured by using a 20x/0.40NA objective lens. The probe beam power was kept sufficiently low (0.6 mW) to avoid self-heating effects in graphene.

Graphene has fundamental D ("disorder") and G ("graphite") peaks located at ~1350 cm$^{-1}$ and 1582 cm$^{-1}$, respectively [21]. In addition, the 2D mode is located at ~2680 cm$^{-1}$ in the case of a pristine monolayer graphene. In the presence of doping or strain, G and 2D peaks are shifted towards 1600 and 2700 cm$^{-1}$, respectively [21,22].

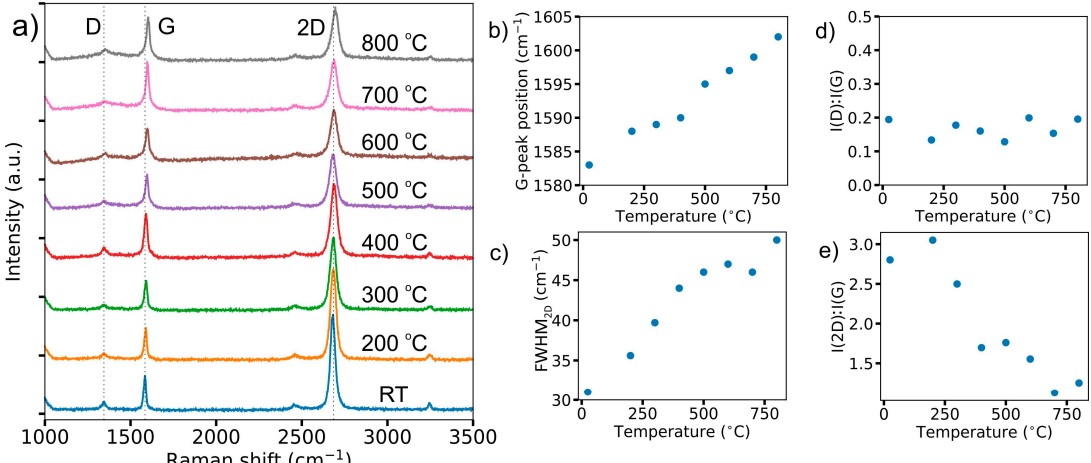

**Figure 3.** Raman characterization. (**a**) Raman spectra of the graphene sample before and after annealing at different temperatures. (**b**) The G-peak position and (**c**) 2D peak width increased linearly. (**d**) The ratio of I(D):I(G) did not depend on the annealing temperature, while (**e**) I(2D):I(G) rapidly decreased when the sample was annealed at temperatures above 300 °C.

Figure 3a shows the Raman spectra of the graphene sample before and after annealing. Even though the temperature increased from room temperature (RT) to 800 °C, there were only minor changes in the spectra. By analyzing the temperature dependence of the position of the G peak, the full width at half maximum of peak 2D (FWHM$_{2D}$), and from the intensity relations of peaks D, G, and 2D, one can observe details that indicate changes in graphene.

We made a Lorentzian fitting for peaks D, G, and 2D, and observed that the position of peak G was located at 1583 cm$^{-1}$ before annealing. Figure 3b shows that the G peak shifted linearly toward 1600 cm$^{-1}$ when the annealing temperature increased. For the sample annealed at 800 °C, the G peak appeared at 1602 cm$^{-1}$. Moreover, Figure 3c shows a linear increase in FWHM$_{2D}$ from 30 to 50 cm$^{-1}$, which was directly proportional to the annealing temperature.

Peak D only went through minor changes, despite high annealing temperatures. This indicates that only a very minor amount of PMMA remained on the graphene after the transfer process [15]. The intensity ratio of D and G peaks remained constant at all annealing temperatures, as is shown in Figure 3d. However, the intensity ratio of the 2D and G peaks decreased from three to one. The decrease in I(2D):I(G) was especially noteworthy in the range of $300-400$ °C. Moreover, we observed a shift in the position of the 2D mode from 2680 cm$^{-1}$ at room temperature to 2694 cm$^{-1}$ after annealing in 800 °C.

## 3. Discussion

The sizes of the Ni contacts in this study were relatively large in comparison to those that are commonly used for the contact resistance measurement [2–6,9–11]. However, it is worth noting that, in our experiment, we measured the same position several times. Since the measurement probe consumed a thin Ni layer by scratching, the surface area of the contact was chosen to be large enough to enable several repetitive measurements. As can be seen in Figure 2c, the annealing of the largest (d2 and d3) contacts had only a minor effect on Rc. However, the importance of the annealing procedure is pronounced when the contact area is small, as has been shown in earlier studies [8,10,12] and in Figure 2c (d1 contact).

Due to the relatively large size of contacts, it is difficult to make a precise estimation of the contact resistance behavior. However, the results clearly show that the annealing of a graphene sample with nickel contacts at temperatures above 600 °C severely damaged the sample by destroying the electrical contact between graphene and nickel. The mechanism behind the damage may originate from high carbon solubility in nickel at elevated temperatures [23]. More precisely, nickel absorbs carbon atoms, and when the graphene layer is fully absorbed, the nickel–graphene contact disappears. This conclusion is supported by an earlier experiment which demonstrated that, at elevated temperatures, Ni can easily break down sp$^3$ hybridization, which has an even higher bonding energy in comparison to that of sp$^2$ hybridized graphene [24–26].

Moreover, it is worth noting that, despite a comparatively high melting temperature of the bulk Ni equal to 1455 °C, the surface melting temperature—the so-called Tamman temperature—is approximately half of this bulk melting temperature [27,28]. At this temperature, nickel atoms acquire enough energy to become mobile [27–30]. For low-dimensional systems (e.g., for contacts that are a few tens of nanometers thick), this temperature is sufficient to cause physical changes [29,30]. For instance, in Reference [30], a 10-nm-thick Ni film was melted at $700-800$ °C, and one may therefore expect that thin Ni contacts will undergo major changes at temperatures above 700 °C.

Figure 2b shows that the un-annealed sample had lower Rg compared to that after annealing at 200 °C. This indicates that the sample surface was affected by moisture in the surrounding air that landed on the sample. We confirmed this by the sample that was first annealed at 300 °C and left for 12 h in air (the relative humidity was ~40%). After waiting, the Rg decreased by almost 40% (the average Rg after the annealing and after 12 h was 521 ± 45 Ω and 327 ± 24 Ω, respectively). This was caused by H$_2$O and O$_2$ doping, which is pronounced after a graphene sample is annealed at high temperatures [16]. However, we also observed that the moisture decreased the Rc value, which suggests that the moisture at the contact point improves the electrical contact between nickel and graphene. Therefore, in Figure 2c one can observe an increase in Rc value before and after the annealing of the sample at 200 °C. When the contacts are deposited on graphene, the effect of the moisture on graphene may have an impact on the resistance between the graphene film and the metal contact. Understanding how moisture affects the contact electrodes will be part of our future work.

The position of peak G shifts toward 1600 cm$^{-1}$ when graphene becomes p-doped or when it is strained [22,31,32]. In addition, a drastic decrease in I(2D):I(G) ratio (Figure 3e) points to the dominant doping effect [31,32]. By measuring the resistance of graphene, we observed that the annealing led to a decrease in Rg, while the strain in the sample should increase the resistivity of graphene [33]. Therefore, one may expect that the shift of the G peak mainly originated from the doping effect in

graphene. However, Figure 2b shows that the resistance of graphene saturated at temperatures above 500 °C to approximately 250 Ω, which suggests simultaneous doping and straining of the graphene.

## 4. Conclusions

We observed that annealing of a graphene sample at temperatures above 200−300 °C may cause an unintentional p-type doping. When the annealing was performed at temperatures above 600 °C, the annealing process became harmful to nickel–graphene contacts. Moreover, in these cases, when the size of the electrical contacts are on the order of millimeters, annealing of a graphene sample with Ni contacts may provide no significant benefits, but in contrast, may increase the doping level in graphene. However, if the doping of a graphene sample is required, then high-temperature annealing provides a simple route for this purpose.

**Author Contributions:** Conceptualization, T.K. and I.K.; methodology, T.K., V.J., A.B., A.L. and G.N.; formal analysis, T.K., G.N. and I.K.; investigation, T.K.; resources, I.K.; writing—original draft preparation, T.K.; writing—review and editing, T.K., V.J., A.S. and I.K.; visualization, T.K.; supervision, I.K.; project administration, I.K.; and funding acquisition, I.K.

**Funding:** This work was supported by the Research Council of Lithuania under the KOTERA-PLAZA project (grant no. DOTSUT-247).

**Acknowledgments:** The authors acknowledge Irena Mirviene for language editing of the manuscript.

**Conflicts of Interest:** The authors declare no conflict of interest.

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
