# Peer review of "Effect of High-Temperature Annealing on Graphene with Nickel Contacts"

_condensedmatter, doi:10.3390/condmat4010021_

Round 1

Reviewer 1 Report

The authors describe the effect of high-temperature annealing on graphene/nickel contacts.  My biggest concern is regarding the novelty of the article which I regard as low due to the existence of several articles describing the effect of annealing on graphene contacts. I would advise the authors to elaborate on the novelty of their findings and the impact of their work. In addition, the discussion of their findings is rather superficial. Therefore I do not recommend publication of the article in its present form.

How does moisture improve the electrical contact? The authors suggest that the graphene is doped but the discussion is rather superficial. A detailed account of water intercalation and its effect on graphene can be found here [https://doi.org/10.1016/j.surfrep.2018.09.001].

Can the authors provide with I(V) curves before and after moisture exposure. This can send light to what kind of doping or which mechanism leads to the reduced contact resistance. 

The authors should provide I(V) curves recorded at each annealing step, recorded with 2 and 4 probe measurements. This could provide additional information on the possible mechanisms contributing to the contact resistance

The graphene doping should be also related to changes in the Raman spectra, see 10.1021/nl2034317. Can the authors discuss this effect? Do they see such changes in their spectra?

I do not understand how strain will eliminate the doping effect at the contacts.

Why there was an atmosphere change cooling from 200C to 100C? 

The manuscript suffers from a misuse of articles (the, a), which significantly reduces its readability. 

Author Response

We appreciate the Reviewers for their time and thank both of them for their useful comments and suggestions that all helped us to improve the manuscript. Below please find our point to point answers to the Reviewers questions. Changes in the revised manuscript are highlighted as requested by the Editor. Moreover, we polished written English as well.

Reviewer #1:

Comment: The authors describe the effect of high-temperature annealing on graphene/nickel contacts.  My biggest concern is regarding the novelty of the article which I regard as low due to the existence of several articles describing the effect of annealing on graphene contacts. I would advise the authors to elaborate on the novelty of their findings and the impact of their work. In addition, the discussion of their findings is rather superficial. Therefore I do not recommend publication of the article in its present form.

Answer: We agree with Reviewer that there exist several works that investigate problems of graphene annealing with and without electric contacts. Most relevant of these papers have been properly cited in the manuscript [Ref.1-12]. However, to our knowledge there are very limited number of works that are dedicated on graphene annealing in extreme temperature range (above 400 °C) and none (that we are aware of) is dedicated on systematical study of graphene annealing in high temperature range with contacts. Specifically, our work demonstrate that Ni-graphene contacts start degrade after 600 °C and are fully destroyed at 700 °C, while graphene survives even 800 °C temperature. We kindly notify that such information may prove to be useful for those who consider processing of the graphene in high temperature conditions. Therefore, we trust that our work brings new data into the field.

Following by the Reviewer’s suggestions, we extended the introduction in the revised manuscript. Also, the discussion was carefully revised to provide deeper understanding especially on the doping mechanism. Moreover, the novelty of the results is explained more carefully for clarifying the aim of our work.

Comment: How does moisture improve the electrical contact? The authors suggest that the graphene is doped but the discussion is rather superficial. A detailed account of water intercalation and its effect on graphene can be found here [https://doi.org/10.1016/j.surfrep.2018.09.001].

Answer: This is very interesting question and we thank Reviewer for providing a helpful paper. The mechanism how moisture and air (H2O and O2) dopes graphene after annealing is discussed here (https://doi.org/10.1002/jrs.2485 – Ref. [16] in the revised manuscript). Nevertheless, the precise mechanism why moisture decrease resistance at contacts is yet unclear. One possible mechanism would be that the moisture may accumulate to the contact point of Ni and graphene thus improving the electron transport between graphene and nickel.

Although the topic is interesting, we should kindly remind that the aim of our work is to study graphene and Ni-graphene contacts at elevated temperatures. Studying effect of moisture on graphene and Ni-graphene contacts will be topic of our further work. However, to expand the discussion, we have now briefly discussed H2O/O2 doping mechanism at page 5.

Comment: Can the authors provide with I(V) curves before and after moisture exposure. This can send light to what kind of doping or which mechanism leads to the reduced contact resistance. 

Answer: We agree with the Reviewer, that this topic should be studied more carefully. However, it goes rather far beyond from the aim of this work and therefore, as we state in our manuscript (at page 5), this topic will be studied in the near future.

Comment: The authors should provide I(V) curves recorded at each annealing step, recorded with 2 and 4 probe measurements. This could provide additional information on the possible mechanisms contributing to the contact resistance

Answer: We thank reviewer for this comment. The I(V) curve of 2-contact measurement is indeed often used to measure Ohmic or Schottky behavior of the graphene-metal contacts. In several works have been demonstrated that Ni-graphene contacts are Ohmic. We cited one of these works by adding the Ref. [7] in the revised manuscript. To confirm that this applies to our contacts as well, we measured the I(V) curve of a sample with Ni-graphene contacts. The results are shown in Fig. C1. The I(V) curve demonstrates a linear behavior (i.e. the contacts are Ohmic). Moreover, there are no reason to expect that the Schottky barrier would appear during annealing, especially since the resistance of the contacts decreases as function of annealing temperature before Ni-graphene contacts break down.

Figure C1. Linear I(V) curve shows that there is no Schottky barrier between Ni and graphene.

Concerning the I(V) curves measured by the 4-point-probe technique, we respectfully disagree with the Reviewer, that this could provide additional information on the possible mechanism contributing to the contact resistance. The 4-point-probe technique is used to eliminate the contact resistance and measure only the resistance of graphene. Furthermore, since the graphene is gapless semimetal, there are no reason to believe that I(V) curves measured by 4-point-probe technique would provide additional information before or after annealing (i.e. I/V characteristics of graphene are linear before and after doping).

Therefore, in summary, despite we appreciate the Reviewer’s comments, we decided to not include I(V) curves in the revised manuscript. However, we included the comment about Ohmic behavior of the fabricated contacts (page 2 last paragraph).

Comment: The graphene doping should be also related to changes in the Raman spectra, see 10.1021/nl2034317. Can the authors discuss this effect? Do they see such changes in their spectra?

Answer: We appreciate this helpful comment from Reviewer. The paper presented by Reviewer will surely help us to understand the mechanism of doping caused by the moisture. Moreover, the effect of H2O/O2 doping after annealing is discussed here (https://doi.org/10.1002/jrs.2485) and we believe this is the main mechanism in our case. We cited the paper (Ref. [16]) and briefly discussed this at page 5 in the revised manuscript.

Comment: I do not understand how strain will eliminate the doping effect at the contacts.

Answer: We are thankful for this comment. The sentence (last sentence in discussion section) was poorly written and thus misleading. We expect no strain in the contacts but strain in graphene. Strain in graphene increase resistance and while doping decrease resistance. Therefore, due the strain the Rg stays the same despite doping increases. To clarify our message, we have carefully revised this sentence (at page 5).

Comment: Why there was an atmosphere change cooling from 200C to 100C? 

Answer: We appreciate this observation from the reviewer. The thermal annealing system cools down rather slowly after 200 C. To increase the cooling of the chamber we made the cooling from 200 C to 100 C in N2 atmosphere. This is now explained properly in the text (page 3).

Comment: The manuscript suffers from a misuse of articles (the, a), which significantly reduces its readability. 

Answer: We carefully considered the use of articles and also asked professional English reader to improve readability of the text.

Reviewer 2 Report

The manuscript entitled "Effect of High Temperature Annealing to Graphene with Nickel Contacts" by Kaplas et al. reports changes in resistance and Raman peaks of graphene with nickel contacts on a SiO2/Si substrate by annealing. The authors fabricated the samples with and without the annealing process at temperatures in the range from 200 to 800 °C and conducted transport measurements based on two- and four-point probe methods and Raman scattering measurements for them. The paper proposes possibilities of (1) unintentional doping to the graphene with the annealing temperature above 200-300 °C and (2) damage of the electrical contact between nickel and graphene with the annealing temperature above 600 °C, and it might give important information in the field of nanoscience and nanotechnology. Thus, I think the paper deserves publication in Condensed Matter. However, I recommend the revision concerning the following points for improvement of the paper.

1) The 2D-peak width increases with increase in the annealing temperature. How is it explained according to the authors’ claim? When the number of layers of graphene increases, the 2D-peak width increases and I(2D)/I(G) ratio decreases. These spectral features were also observed in this study. Is there any possibility of increase in the number of layers of graphene with nickel by the annealing process?

2) line 184: Figure 2(b) shows that the unannealed sample has lower (not higher) Rg than that after the annealing at 200 °C.

Author Response

We appreciate the Reviewers for their time and thank both of them for their useful comments and suggestions that all helped us to improve the manuscript. Below please find our point to point answers to the Reviewers questions. Changes in the revised manuscript are highlighted as requested by the Editor. Moreover, we polished written English as well.

Reviewer #2:

Comment: The manuscript entitled "Effect of High Temperature Annealing to Graphene with Nickel Contacts" by Kaplas et al. reports changes in resistance and Raman peaks of graphene with nickel contacts on a SiO2/Si substrate by annealing. The authors fabricated the samples with and without the annealing process at temperatures in the range from 200 to 800 °C and conducted transport measurements based on two- and four-point probe methods and Raman scattering measurements for them. The paper proposes possibilities of (1) unintentional doping to the graphene with the annealing temperature above 200-300 °C and (2) damage of the electrical contact between nickel and graphene with the annealing temperature above 600 °C, and it might give important information in the field of nanoscience and nanotechnology. Thus, I think the paper deserves publication in Condensed Matter. However, I recommend the revision concerning the following points for improvement of the paper.

 Answer: We thank Reviewer for his/her kind comments.

Comment: The 2D-peak width increases with increase in the annealing temperature. How is it explained according to the authors’ claim? When the number of layers of graphene increases, the 2D-peak width increases and I(2D)/I(G) ratio decreases. These spectral features were also observed in this study. Is there any possibility of increase in the number of layers of graphene with nickel by the annealing process?

Answer: We thank reviewer for this interesting question. It is unlikely that the number of layers in graphene increases during annealing because the thermal annealing does not involve carbon precursor. However, the 2D peak widening and decrease of I(2D)/I(G) ratio originate from p-doping effect caused by H2O/O2 reactions after annealing (https://doi.org/10.1002/jrs.2485). This observation by electrostatic doping was done by A. Das et al. in Nat. Nanotech 2008, 3, 210-125. To extent the discussion, we have included a brief explanation to this effect.

Comment: line 184: Figure 2(b) shows that the unannealed sample has lower (not higher) Rg than that after the annealing at 200 °C.

Answer: We are thankful for the reviewer for noticing this mistake. It is corrected in the revised manuscript version. 

Once again, the authors thank the Reviewers for careful reading and comments that all helped us to improve the manuscript.

Round 2

Reviewer 1 Report

The authors have adequately answered most of my questions. The paper is now publishable.